https://doi.org/10.1038/s42003-021-02946-w — OPEN
# Inferring the stabilization effects of SARS-CoV-2 variants on the binding with ACE2 receptor

Mattia Miotto [1], Lorenzo Di Rienzo[1], Giorgio Gosti [1], Leonardo Bo'[1], Giacomo Parisi[1], Roberta Piacentini [1,2], Alberto Boffi [1,2], Giancarlo Ruocco [1,3] & Edoardo Milanetti [1,3 ✉]

As the SARS-CoV-2 (severe acute respiratory syndrome coronavirus 2) pandemic continues to spread, several variants of the virus, with mutations distributed all over the viral genome, are emerging. While most of the variants present mutations having little to no effects at the phenotypic level, some of these variants are spreading at a rate that suggests they may present a selective advantage. In particular, these rapidly spreading variants present specific mutations on the spike protein. These observations call for an urgent need to characterize the effects of these variants' mutations on phenotype features like contagiousness and anti-genicity. With this aim, we performed molecular dynamics simulations on a selected set of possible spike variants in order to assess the stabilizing effect of particular amino acid substitutions on the molecular complex. We specifically focused on the mutations that are both characteristic of the top three most worrying variants at the moment, i.e the English, South African, and Amazonian ones, and that occur at the molecular interface between SARS-CoV-2 spike protein and its human ACE2 receptor. We characterize these variants' effect in terms of (i) residue mobility, (ii) compactness, studying the network of interactions at the interface, and (iii) variation of shape complementarity via expanding the molecular surfaces in the Zernike basis. Overall, our analyses highlighted greater stability of the three variant complexes with respect to both the wild type and two negative control systems, especially for the English and Amazonian variants. In addition, in the three variants, we investigate the effects a not-yet observed mutation in position 501 could provoke on complex stability. We found that a phenylalanine mutation behaves similarly to the English variant and may cooperate in further increasing the stability of the South African one, hinting at the need for careful surveillance for the emergence of these mutations in the population. Ultimately, we show that the proposed observables describe key features for the stability of the ACE2-spike complex and can help to monitor further possible spike variants.

[1] Center for Life Nano & Neuroscience, Istituto Italiano di Tecnologia, Viale Regina Elena 291, 00161 Rome, Italy. [2] Department of Biochemical Sciences "Alessandro Rossi Fanelli", Sapienza University of Rome, P.Le A. Moro 5, 00185 Rome, Italy. [3] Department of Physics, Sapienza University, Piazzale Aldo Moro 5, 00185 Rome, Italy. ✉email: edoardo.milanetti@uniroma1.it

The severe acute respiratory syndrome coronavirus 2 (SARS-CoV-2) infection was firstly observed in late 2019[1,2]. In the subsequent months of epidemic spreading, many SARS-CoV-2 variants, viral sequences characterized by at least one mutation with respect to the original one, have been detected worldwide[3].

In coronaviruses, mutations naturally occur during viral replication, and despite the fact that coronaviruses encode for an enzyme that corrects the errors, some of these mutations are preserved, originating new variants. As for all biological systems, the action of natural selection eventually tends to fix in the genome the mutations characteristic of those variants that present an increase of the fitness, and it has been registered that in these months the rate of emergence of new SARS-CoV-2 variants is about two variants per month[4,5].

This rapid proliferation of variants poses a further threat for the community as the virus can acquire different phenotypes. For instance, the main mutation of the B line, involving the amino acid substitution D614G in the spike protein, is established since March 2020 and it is now largely dominant in patients[6–8]. This mutation would allow the receptor-binding domain (RBD) of the spike to assume a conformation more suitable to bind Angiotensin-converting enzyme 2 (ACE2), and it could be responsible for the increased viral action[8,9].

More recently, on December 14, 2020, authorities in the United Kingdom of Great Britain and Northern Ireland reported to World Health Organization that a new SARS-CoV-2 variant, named B.1.1.7, and commonly known as the English variant, has been identified via viral genomic sequencing[10,11]. The variant is defined by the presence of a range of 14 mutations involving amino acid modifications and three deletions, including the spike D614G mutation. Even if investigations are underway to determine whether this variant is associated with any changes in antibody response or vaccine efficacy, it seems to be characterized by an increased transmissibility[12] and lethality[13], also because it is spreading with very high speed all over the world.

Among the mutations affecting the spike protein, like 69-70del or P681H, mutation N501Y is located in the region that directly contacts the ACE2 receptor. Therefore, it is possible that this mutation could have a direct effect on the binding affinity between the two proteins[14].

Furthermore, the importance of this amino acid substitution is confirmed by its presence in two other rapidly spreading variants, B.1.351[15] and P.1[16] (commonly referred to as South African and Amazonian variants, respectively).

These two variants both include other amino acid substitutions in the spike binding site, making clear that this region is under severe evolutionary pressure. Indeed, selectivity and affinity of the spike toward its main receptor, ACE2, remain the crucial factors determining SARS-CoV-2 contagiousness and virulence.

Even if, it has been demonstrated that SARS-CoV-2, similarly to other coronaviruses[17,18], is able to bind to sialoglycan-based receptors in the spike's N-terminal domain[19–21].

From a molecular point of view, the spike protein of CoVs, protruding from the viral membrane, not only plays a crucial role as a fundamental structural protein, but it also is essential for the interaction between CoV systems and host cells[22]. Structurally, the spike protein is found in the trimeric complex, each chain composed of two sub-units: S1 and S2. The Receptor Binding Domain (RBD), located in the S1 domain, is responsible for viruses' interaction with receptors on the host cell surface[23]. On the other hand, the S2 subunit is responsible for the fusion between the virus and host membrane, causing the viral genome to penetrate the host cell's cytoplasm[24].

Interestingly, the interaction with ACE2 involves the C-terminal domain of SARS-CoV-2 spike protein, whose amino acid sequence is well conserved with respect to SARS-CoV homologous one[25]. Conversely, the N-terminal domain presents some insertions, and these additional surface regions could be used by the virus to bind other cell receptors, so constituting an additional cell entry mechanism[26].

Here, we perform a set of molecular dynamics simulations on different spike-receptor complexes, involving mutations at the spike protein interface. In particular, we both consider single mutations belonging to the binding site of the spike protein for which the binding affinity experimental data is known[27], and some of the variants of SARS-CoV-2 are currently most widespread in the world. We analyze the mutations for which binding affinity is known in order to detect the dynamic-structural properties of the ACE2-spike complex that give the spike protein a greater propensity to interact with the molecular partner. Only by considering the RBD of the spike protein and the extra-membrane domain of the ACE2 receptor, we analyze the dynamic properties of the residues belonging to the two interfaces. Furthermore, an analysis based on graph theory was performed in order to investigate the stability of the contacts during molecular dynamics simulations. Finally, we analyze the shape complementarity of the two interfaces over time, using Zernike polynomials to characterize the shape of each portion of the molecular surface, establishing a complementarity value between each pair of surface[28–31]. Also, in this case, we investigate the stability of the geometrical matching between the two interfaces during the molecular dynamics simulation because stable binding to the host receptor plays a crucial role for virus entry mechanisms[32].

Overall, our study shows that the English variant and the South African variant have structural and dynamic properties similar to the N501F single-mutation variant, which is experimentally known to be the one with the highest binding affinity with ACE2[27]. This finding invites us to explore the possible cooperative effect of mutations in the binding region. Therefore, we substituted the F amino acid (Phenylalanine) at position 501 of the spike protein for both the Amazonian and South African variants. The results obtained from our fully computational approach show that the presence of the F amino acid in the Amazonian variant would worsen the binding affinity. On the other hand, the same mutation carried out in addition to the mutations of the South African variant would increase the stability of the spike-ACE2 complex, suggesting these mutations as a possible worrying variant.

## Results and discussion

**The role of spike mutations in the emerging variants**. Although mutations are showing up all over the SARS-CoV-2 viral genome, those taking place on the spike protein are under intense scrutiny as they are expected to directly impact viral entrance in the host cells, thus on transmissibility and infectivity. The first notable mutation that has been observed involved residue 614 which changed from D to G[6]. This mutation even if localized in a region distant from the binding site, is rapidly fixed in the population suggesting an indirect effect on the phenotype. Indeed, a recent computation study highlighted a conformational change driven by such mutation that favors ACE2 receptor binding, thus explaining the phenotypic advantage[8].

With the huge spread of the epidemics, other mutations accumulated over the SARS-CoV-2 genome. In particular, a variant of the virus with six mutated amino acids and two deletions in the spike protein emerged in England in late 2020. The English variant has the D614G mutation together with mutations/deletions belonging to spike protein: 69–70 HV deletion, 144Y deletion, N501Y, A570D, P681H, T716I, S982A,

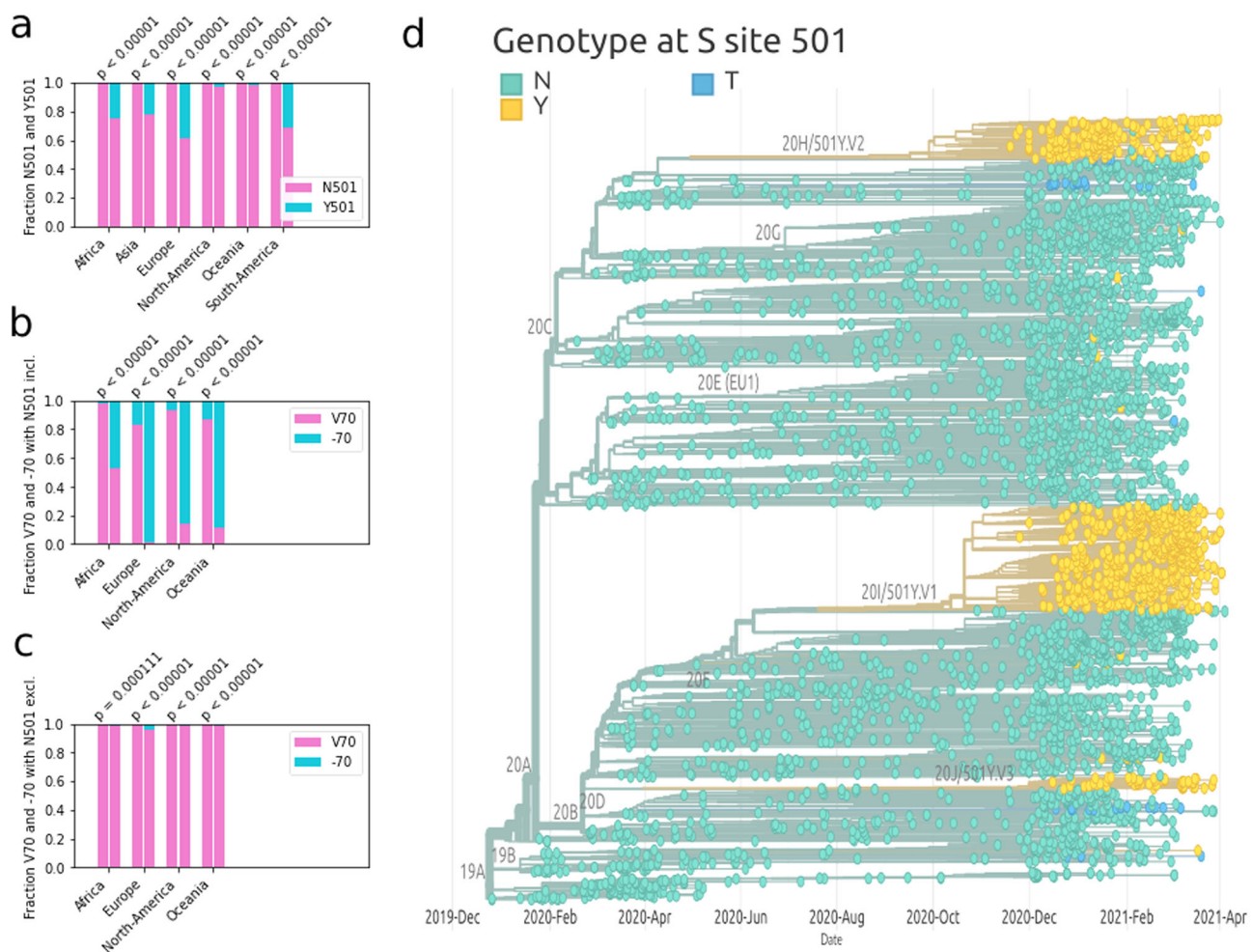

**Fig. 1 Sequence analysis of the English variant mutations. a** Fractions of sequenced viral samples having amino acid N in position 501 of the spike protein (orange) vs those having Y (blue) for two different time intervals and in different countries. **b** Same as panel (**a**), but comparing samples with residue V70 (orange) and samples for which deletion of residue 70 took place (blue). **c** Same as in panel (**b**) but considering only sequences where residue 501 is an N. **d** Phylogenetic tree of the SARS-CoV-2 variants. Leaves are colored in light green if the corresponding spike sequence has an N in position 501, in dark blue in case of a T, and in yellow for a Y.

and D1118H. Specifically, mutation N501Y and deletions 69–70 attracted much attention since the mutation falls in the receptor-binding domain (RBD) while the double deletion interests the N-terminal domain, which has been demonstrated to bind sialic acid-rich attachment factors[19,21].

To begin with, we looked at genomic data taken from infected individuals divided by geographical location—which is provided by the regularly updated public website (https://cov.lanl.gov)—to assess the variance invasion potential. In particular, analyses are based on 823,121 spike alignment sequences taken at different times and different geographical areas using the consistency and significance test presented by Korber et al.[33]. First, we compared sequences carrying the "wild type" residue in position 501 (i.e., N501) with the ones carrying the 501Y mutation. Note that, in this work, we considered the spike protein first detected in Wuhan at the beginning of the epidemics as the "wild type" (WT).

Figure 1a displays the changes of relative frequencies for N501 and Y501 for two-time points separated by 2 weeks. The first time point represents all sequences up to the onset day, and the second time point includes all the sequences acquired at least 2 weeks after the onset date (at least 15 sequences are required for significance). One can see that in each country, the variant 501Y is rapidly spreading.

The significance of the signal is obtained by testing the null hypothesis that the mutation does not affect the variant fitness and that consequently, the relative frequency shift must randomly go with equal probability in each direction. The $p$ values confirm that in all continents we observe a systematic increase of the relative frequency of Y501. Next, we moved to consider sequences carrying the deletion V70 (taking place on the glycan-binding domain). Figure 1b, c show respectively the change in the relative frequency of the deletion at position 70 constraining respectively for the N501 and Y501 variant; this shows that the deletion of the 70th position increases in relative frequencies at a higher rate in the Y501 variant. This suggests that deletion 70 is driven by the rapid spread of the English variant, and/or that the Y501 variant gives a stronger relative change in fitness with respect to the deletion.

Finally, looking at all infection data, we compare the invasive potential of the different 501 mutations. The phylogenetic tree in Fig. 1d was obtained using the publicly available interactive visualization platform Nextstrain[34] and it is based on the contribution of 4025 genomes sampled between December 2019 and April 2021, stored and elaborated in the Nextstation database and bioinformatics pipeline for phylodynamics analysis[34]. A full list of the sequence authors is available at https://nextstrain.org/sars-cov-2. Light green dots represent sampled genomes with

**Table 1 Variants considered in the present work.**

| | |
|---|---|
| Wild type (WT) | N501N |
| English (UK) | N501Y |
| SARS-03 like | N501T |
| Negative control 1 | N501K |
| Negative control 2 | N501D |
| Positive control 1 | N501F |
| South African | K417N-E484K-N501Y |
| Amazonian | K417T-E484K-N501Y |

N501 sequences, yellow dots Y501 and blue T501. One can see that the N501Y mutation independently emerged in different branches of the phylogenetic tree and that in each case it rapidly spreads. Furthermore, up to now the N501T mutation emerged and spread only from a single branch.

Interestingly, experimental measurements of binding affinity upon single mutations found that changing residue 501 from N to both T and Y results in an increase of the affinity, while most of the other possible mutations lead to its decrease[27].

In the following, we propose to investigate in greater detail what are the molecular features responsible for the increase/decrease of complex stability upon mutations.

**Mutational protocol.** In order to analyze the role of amino acid substitutions on the binding between the RBD of the SARS-CoV-2 spike protein and the ACE2 receptor, we performed molecular dynamics simulations on a set of spike variants in complex with human ACE2 and characterized the effects of the different mutations on the stability of the protein complex. Starting from the structural complex comprising the RBD of the WT spike and the human ACE2 receptor (PDB id: 6M0J), we obtained the RBD of variant B.1.1.7 (the English variant) mutating residue 501 from amino acid N to Y.

In addition, we also considered four other single-mutated complexes: N501T and N501F, which are expected to display higher affinity; N501D and N501K, which should exhibit lower stability with respect to the wild type one[27].

Then we move to consider cases in which three mutations are present in the RBD, i.e. we analyze the South African and Amazonian variants carrying the mutations K417N-E484K-N501Y and K417T-E484K-N501Y, respectively. All considered systems are reported in Table 1. As experimentally-resolved structures are not available for all the considered systems, mutations were obtained using the PyMol software[35] (see Methods) and a 500 ns-long molecular dynamics simulation was conducted to equilibrate each system. To validate the mutational procedure, we used available experimental structures, performing two different analyses. First, we retrieved the experimental complexes of the SARS-CoV-2's RBD bound to human ACE2 for the English (PDB id: 7MJN) and Amazonian (PDB id: 7NXC) variants. Then, we carried two additional 500 ns-long molecular dynamics simulations and verify that the configurations explored by the two simulations overlap with those sampled during the MD of the computationally mutated ones. In practice, we compared the structures obtained for both simulations through a principal component analysis (PCA). Indeed, a set of configurations sampled from the simulation of the experimental structure was projected into the essential space, defined by the two principal components of the covariance matrix obtained from the trajectory of the computationally mutated complexes (see Supplementary Figs. 4, 5). Figure 2a, b shows an overlap between the two sets of structures, which indeed have a high degree of similarity in terms of the backbone conformation.

Next, we considered the South African variant's complex, whose experimental structure is not available. In this case, we adopted a different test strategy, starting from the structure of the spike in the unbound form of both the wild type (PDB id:7KJ5) and South African variant (PDB id: 7LYN). In particular, we performed three additional, 500 ns-long molecular dynamics simulations starting from (i) the experimental APO form of the RBD of the South African variant; (ii) a computationally-obtained APO conformation extracted from the experimental WT APO conformation of the spike trimer, computationally mutated into the South African variant, and (iii) the spike RBD bound conformation (obtained from the computationally-mutated complex but removing ACE2).

Then, we compared the dynamical behavior of the spike protein in all three cases with respect to the complex simulation. In particular, Fig. 2c shows the distribution of root mean square displacement (RMSD) among a sampled set of complex-spike conformations (purple curve) and between the complex configurations and a set of configurations extracted from each of the three free-spike dynamics. As one can see, all distributions partially overlap in the region around 1.5–2.0 Å, meaning that they can explore a similar configuration space. As we may expect, the mean RMSD progressively increases from the complex distribution to the computationally-mutated apo spike. Complete RMSD data are shown in Supplementary Fig. 1.

Finally, identifying the centroids of the clustering analysis of the sampled configurations, we performed a molecular docking with the ACE2 receptor structure.

According to silhouette analysis (see Supplementary Fig. 2), the frames of the entire trajectory can be grouped into three different groups, the centroids of which are the most representative structures of the entire trajectory. A docking analysis between ACE2 receptor structure and each of the centroid spike configurations was performed using Hdock web-server[36], where the two interacting regions were constrained (see Method section for more details). The analysis did not show important changes for the three analyzed configurations, therefore only the results related to the centroid belonging to the most populated cluster are reported. The 500 structures sampled from the trajectory and the first 100 docking poses were chained together and a Principal Component Analysis was performed. Each structure was then projected onto the essential plane defined by the first two main eigenvectors (whose eigenvalues explain the DD of the total variance). As expected, the structures of the trajectory take place in a restricted portion of the essential plane, while the structures coming from the docking occupy a greater space (see Supplementary Fig. 3). However, some of the docking structures are very similar to the structures sampled during molecular dynamics. To better appreciate this structural similarity, we consider 16 molecular dynamics structures (which are temporally equidistant) and the 16 docking structures closest to the dynamics structures. The contour lines in Fig. 2d show the probability density of finding a structure in a given portion of the essential plain. In fact, as expected, the region is less dense where the docking poses are projected (orange points) with respect to molecular dynamics frames. To better visualize the molecular docking results, we have extracted the two most representative conformations among the docking poses, one closest to the structures sampled in molecular dynamics and one further away from the trajectory frames and closer to the other docking poses. The results, in terms of structural visualization, are shown in Fig. 2e, f. Note that even in the case of the most distant poses (in blue), the binding conformation is very similar to the one most representative of molecular dynamics (in gray).

Overall, all the performed analyses confirmed that the obtained results are not significantly affected by the choice of a different

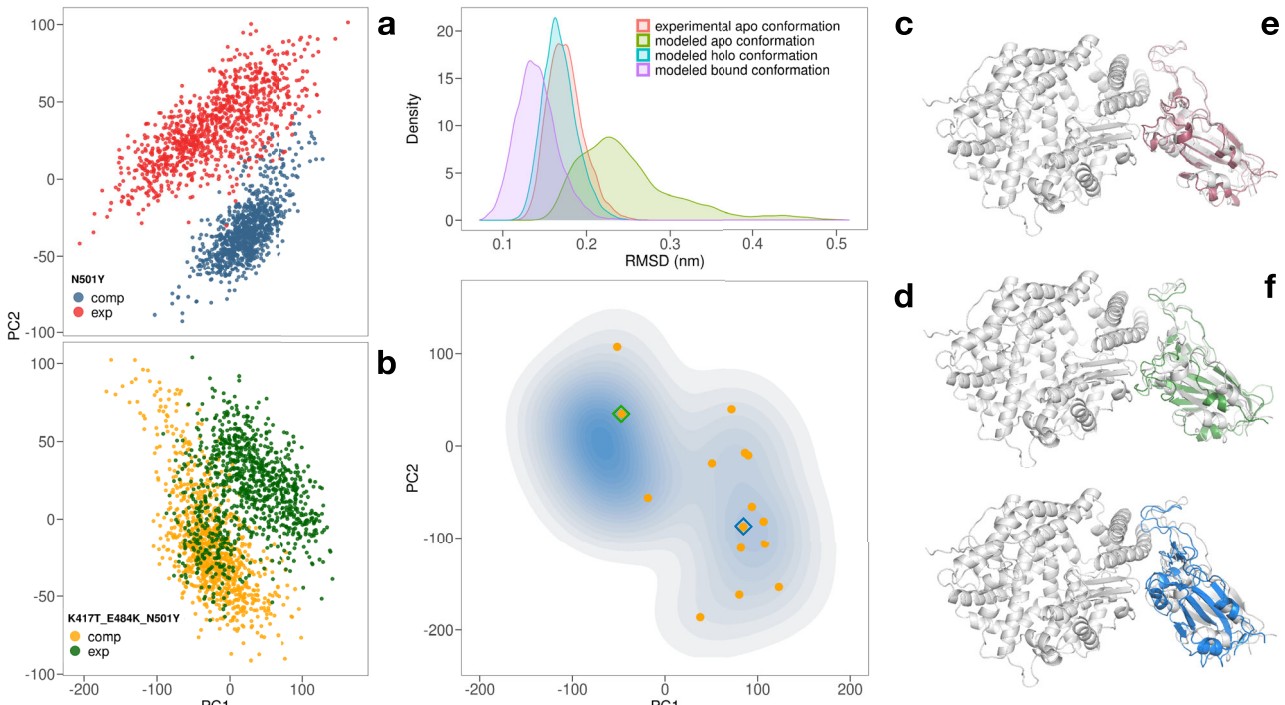

**Fig. 2 Comparison between experimental and modeled SARS-CoV-2 Spike-Ace2 complexes. a** Comparison between the molecular dynamics simulation of the experimental and computationally-obtained complex of the English variant of SARS-CoV-2 spike protein bound to ACE2 receptor. Snapshots of the spike protein are projected in the plane of the two major components of the covariance matrix obtained from a principal component analysis (PCA) over the position covariance matrix. **b** Same as in panel (**a**) but for the SARS-CoV-2's Amazonian variant. **c** Distribution of the root mean square deviation (RMSD)of the spike protein of the SARS-CoV-2 South African variant in apo and holo (bound to ACE2 receptor) forms. Fifty sampled frames of the spike protein from the computationally-obtained complex were used as references to evaluate the RMSD. **d** Probability density of finding a configuration of the apo spike protein in the plane identified by the two principal components of the covariance matrix. Colored dots represent the position of the configuration produced by the docking procedure when projected in the plane. **e** Cartoon representation of SARS-CoV2 spike frame extracted from the apo simulation having smallest RMSD. **f** Cartoon representation of two SARS-CoV2 spike-ACE2 complexes obtained from the docking procedure.

simulation starting point, being it either an experimental structure or a computationally mutated one. Indeed, this is testified both by the comparison between mutated vs experimental complexes of the English and Amazonian variants and by the analyses on the MD of the South African variant's RBD in apo and mutated-holo forms. We might speculate that the peculiar stability of the RBD fold is motivated by the fact that the spike protein has to continuously optimize its binding site to infect different hosts in different conditions; thus it could be optimal to have a stable fold upon single/few point mutations.

To ensure a complete equilibration of the systems, only configurations after 250 ns are used in the analyses.

**Fluctuation of interface residues for the different variants**. The first observable, we consider to evaluate the stability of each RBD-ACE2 complex, was the root mean square fluctuation (RMSF). In Fig. 3a, we show for each residue, both for spike protein and for the ACE2 receptor, the RMSF value obtained considering each molecular dynamics simulation at equilibrium. The average of the RMSF of all contact residues provides information on the overall mobility of the interface. The binding residues of ACE2 and spike have a comparable mean fluctuation value between them (of about 1.25 Å, as the average of all systems). The negative control 1 and 2 systems (whose mutations are N501K and N501D, respectively) are characterized by a higher binding affinity (quantified by the dissociation constant, $K_d$) than the WT form of the spike. This is clearly evident in terms of atomic fluctuation, given that these systems show greater mobility of the interface residues, with an RMSF average of 1.53 Å and 1.31 Å,

respectively. The system characterized only by N501Y mutation (UK variant) is the most stable of all, with an average of 1.01 Å. The Amazonian variant and the South African variant also have a low average fluctuation compared to the other systems, of $1.14 \pm 0.23$ and $1.22 \pm 0.20$ Å, respectively. A more detailed analysis was performed on the atomic fluctuation of residue 501. In all systems, except for the South African variant, residue 501 of the spike protein has a lower average fluctuation than the average of all residues belonging to the same interface. In particular, the English variant, the Amazonian variant, and the positive control mutation (N501F) show a fluctuation of the residue 501 lower than the average of the RMSF values. Indeed, the RMSF value for residue 501 in these three systems is 1.00, 1.00, and 1.20, while the average values of RMSF for the three systems are $1.01 \pm 0.16$, $1.14 \pm 0.23$, and $1.25 \pm 0.20$, respectively. Therefore, for these three systems, residue 501 is particularly stable with respect to the other residues. On the contrary, the other systems have a more pronounced fluctuation of residue 501. Among all, the two negative control systems, N501K and N501D mutations are characterized by a very high average RMSF value: 1.83 and 2.03 respectively. The results of the analysis of atomic fluctuation over time of the interface residues highlight the stability of systems with higher binding affinity, including the South African and Amazonian variants for which we have not considered experimental data (since they are not present in ref. [27]). In order to investigate the stability of intermolecular interactions during molecular dynamics, we calculated the contact frequency matrix for each system. Therefore, we define contact between two residues if their $\alpha$-carbon atoms have a distance less than 9 Å. For

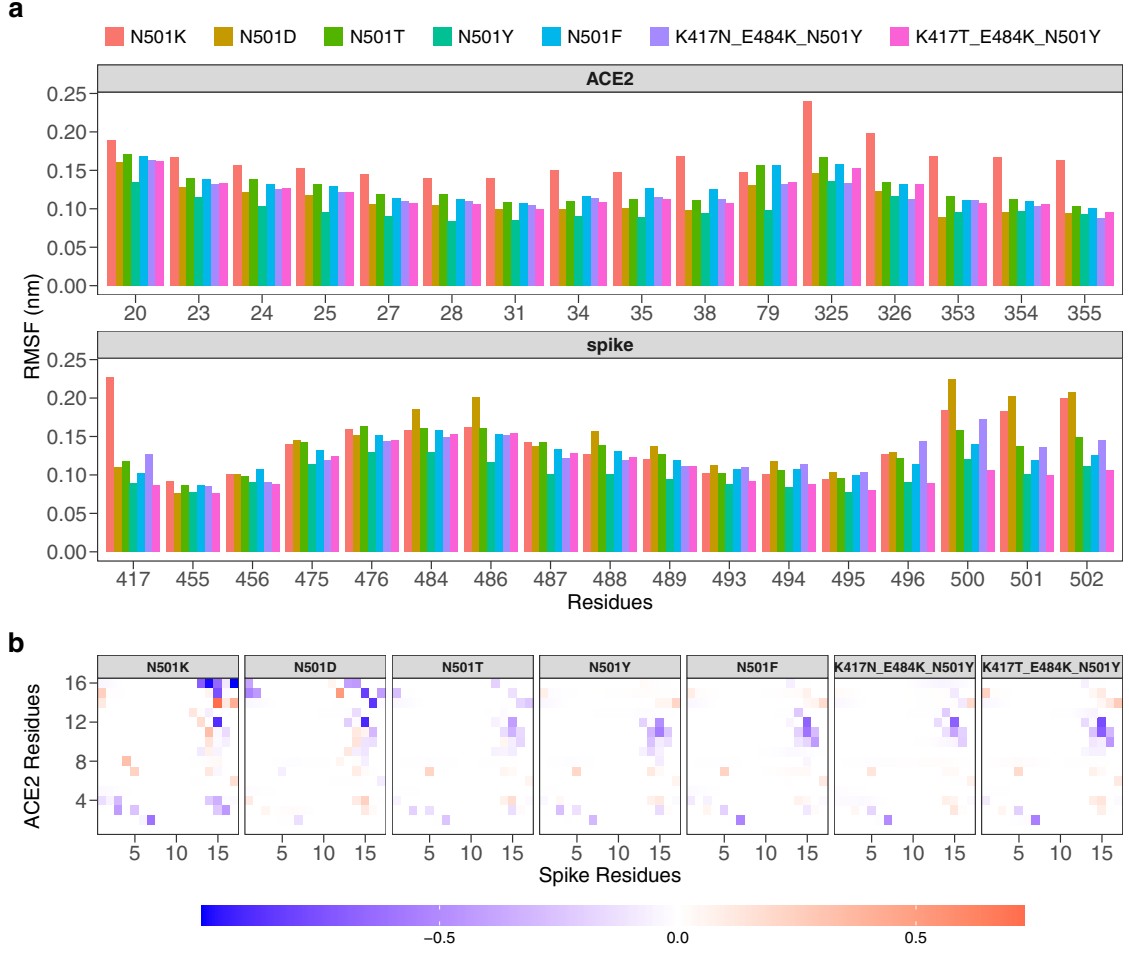

**Fig. 3 Analysis of the motion of the residues of the binding region. a** Root mean square fluctuations (RMSF) of ACE2 (top) and SARS-CoV-2 spike (bottom) protein residues found in interaction during the dynamics. Different colors corresponds to different spike variants (see Table 1). **b** Difference between the contact probability matrices of the interacting residues between each spike variant and the Wuhan (WT) one (see Table 1). Residues are considered as interacting if the distance between their $\alpha$-carbons is lower than 9 Å.

**Table 2 Correlation matrix between couples of variants. Correlation values (resp. *p* values) are reported in the upper (resp. lower) triangular matrix.**

|  | K417N E484K N501Y | K417T E484K N501Y | N501D | N501F | N501K | N501T | N501Y |
|---|---|---|---|---|---|---|---|
| K417N-E484K - N501Y | - | 0.91 | 0.44 | 0.92 | 0.38 | 0.67 | 0.77 |
| K417T-E484K - N501Y | $<10^{-14}$ | - | 0.20 | 0.96 | 0.31 | 0.61 | 0.87 |
| N501D | $3.6 \times 10^{-14}$ | $8.1 \times 10^{-4}$ | - | 0.23 | 0.35 | 0.52 | 0.17 |
| N501F | $<10^{-14}$ | $<10^{-14}$ | $9.8 \times 10^{-5}$ | - | 0.34 | 0.67 | 0.85 |
| N501K | $8.7 \times 10^{-11}$ | $2.6 \times 10^{-7}$ | $4.6 \times 10^{-9}$ | $1.4 \times 10^{-8}$ | - | 0.28 | 0.26 |
| N501T | $<10^{-14}$ | $<10^{-14}$ | $<10^{-14}$ | $<10^{-14}$ | $2.4 \times 10^{-6}$ | - | 0.66 |
| N501Y | $<10^{-14}$ | $<10^{-14}$ | $5.1 \times 10^{-3}$ | $<10^{-14}$ | $9.8 \times 10^{-6}$ | $<10^{-14}$ | - |

each matrix element, we then report the contact frequency between each residue of the spike protein and each other of the ACE2 receptor, then subtracting (to facilitate the comparison) each of these matrices with that obtained for the WT system (see Methods section). As shown in Fig. 3b, we notice similarities between some matrices. For example, the matrix relating to the N501Y system is particularly similar to the N501F system, which is characterized by the highest experimental binding affinity value. An analysis of the Pearson correlation between each pair of matrices allows us to quantify this evidence. In Table 2 the

correlation values between all the contact matrices are shown. Interestingly, the positive control system (characterized by the N501F mutation) has a mean contact map highly correlated with that of the UK, Amazon, and South African variants, with a Pearson correlation value of 0.96, 0.92, and 0.85 respectively. The two systems formed by the two mutations with low $K_d$ (the two negative controls), on the other hand, show lower correlation values with any other system, with an average of the Pearson coefficient of 0.34 and 0.32 respectively. These results represent the first level of classification of the mutated systems of this study

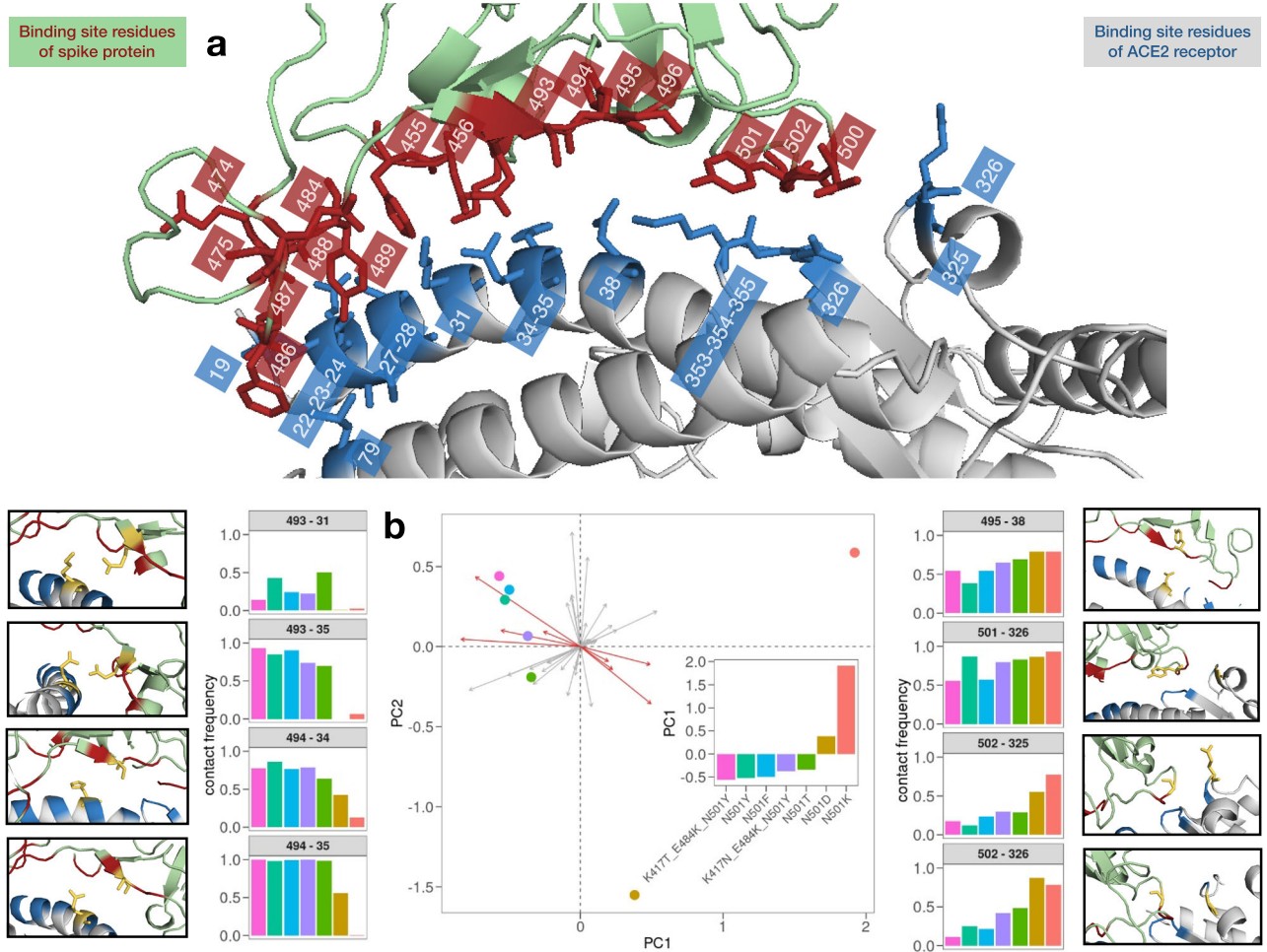

**Fig. 4 Analysis of the covariance in the motion of the interacting residues. a** Ribbon representation of the binding site of the SARS-CoV-2 spike protein bound to human ACE2 (PDB id: 6M0J). Interacting residues, i.e., the ones whose $\alpha$-carbons have a distance lower than 9 Å, are represented with red sticks for the spike protein and with blue ones for the ACE2 receptor. **b** Projections on the two principal components of the analyzed variants (colored dots). Principal component analysis was performed over the contact probabilities of each couple of interacting residues reported in panel (**a**). The inset shows the projections on the first component. Side plots display the contact probability values for the eight couples of residues that contributed the most in differentiating the variants in the principal component analysis.

and allow us to identify the properties of mean atomic fluctuation at a single residue level.

**Principal component analysis of contact frequency**. In order to obtain a more exhaustive overview of the binding between ACE2 and spike protein, we investigate the frequency of the contacts for each analyzed system. To this end, we evaluated the intermolecular contact pairs for each complex, i.e., we consider the percentage of contacts each residue of the spike binding region forms with ACE ones during the simulation. The binding region is composed of 17 residues of the spike protein and 16 belonging to the ACE2 receptor (see Fig. 4a and Methods). Therefore, each system can be described by a vector of 272 contact frequencies. To compactly compare those vectors, we performed a principal component analysis of the vectors' covariance. The projection of each system on the essential plane of the first two principal components (which explain the 54.4 and 35.3% of the total variance, respectively) allows to clearly distinguish, completely unsupervised, the mutations with high affinity from those with low binding affinity. In particular, the projection along the first principal component shows a very interesting trend (as shown in Fig. 4b): the two mutations of the negative control have a

positive value on the first component (distinguishing themselves from any other). On the other hand, the first three systems along this component are the Amazonian variant, the English variant, and the positive control (N501F). We also analyze the loading of each interacting residue pair on the first principal component. In Fig. 4 we show the considered pairs, given that their projection along the axis of the first component is high compared to the others (and the trend of the contact frequency is not trivial between the different systems). In most cases, the pairs of interacting residues at the interface show a trend (increasing or decreasing) of the frequency of the contacts for the different systems. For example, the percentage of contacts during the simulations of the pair composed of the S494 residue of the spike protein and the H34 residue of the ACE2 receptor (S494-H34), progressively decreases for systems with low binding affinity. Therefore, this pair is more stable in high binding affinity systems. With an opposite trend, the pair of residues G502-Q325 shows an increasing trend, meaning that this is more present for systems with low binding affinity. This analysis provides information on which pairs of intermolecular interactions are more stable in high-affinity binding systems and which pairs, on the other hand, are most present in low-affinity binding systems.

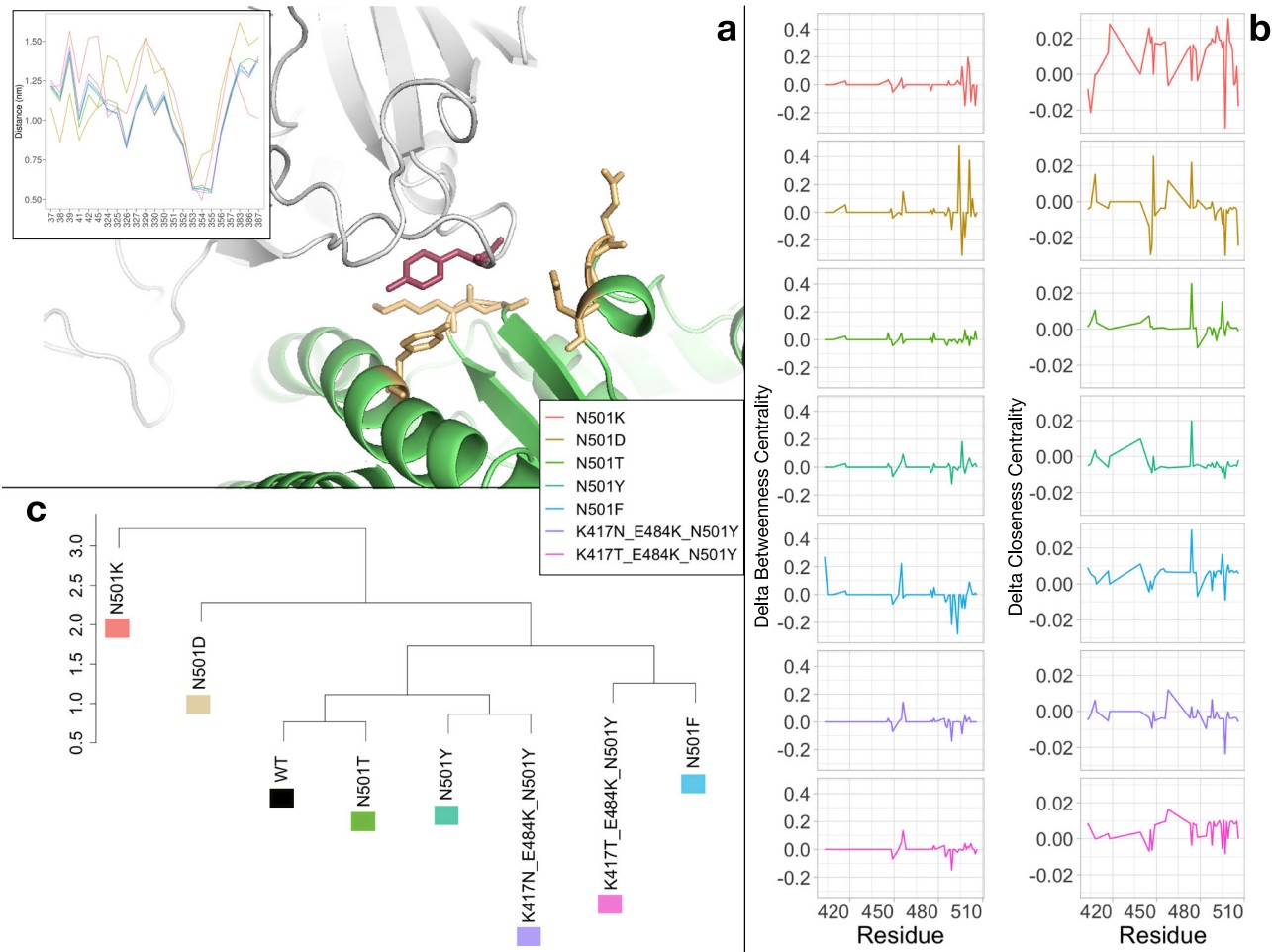

**Fig. 5 Graph analysis of the interacting residues' network. a** Cartoon representation of the complex (PDB id:6M0J) between SARS-CoV-2 spike protein (gray) and the human ACE2 receptor (green). Residue 501 of the spike protein is represented in red sticks, while a set of interacting residues of the ACE protein are identified in yellow sticks on the basis of a spatial distance threshold of 9 Å. The inset shows the average minimum distance of the ACE2 interacting residues with residue 501 of the spike protein for each variant. **b** Difference in Betweenness (left column) and Closeness (right column) centrality of each node of the SARS-CoV-2 spike found in interaction with ACE2 receptor between the seven studied spike variants and the WT one (see Table 1). **c** Hierarchical clustering of the seven spike variants together with the WT reference one (see Table 1).

**Graph theory-based analysis of binding site residues.** A higher level of complexity of the binding properties was addressed by analyzing the organization of interactions between the interface residues. To this end, we model the interaction between the two proteins as a bipartite network, schematizing each residue as a node of the network and each intermolecular interaction as a network edge. Only intermolecular interactions were considered for this analysis, so interactions between two residues belonging to the same protein are not included in the graph definition. In particular, we define a weighted graph for each system, weighing each link connecting two residues with the corresponding contact frequency calculated from the molecular dynamics configurations. In this case, we defined a contact between two residues if their distance is less than 8.5 Å in agreement with Chakrabarty et al. [37], where similar network analyses were performed on protein structures to assess the best threshold for evaluating the centroid-centroid contacts.

As an example for a better interpretation of this modeling for the molecular complex, we report the distances that every residue belonging to the ACE2 binding site has with the residue 501 of the spike protein since this is of primary importance in the variants considered in this work (see Fig. 5a). The mean distance between residue 501 of the spike protein and any other (within 12 Å) of the ACE2 receptor was calculated. This analysis shows

that the systems with lower experimental binding affinity are typically characterized by a greater distance (N501K and N501D single mutation) between the residues belonging to the binding site of ACE2 and the residue 501 of the spike protein. Interestingly, the system with the N501D mutation has a shorter distance than any other system between residues 501 and residues 37, 38, 39, 41, 42, and 45 of the receptor. Similarly, for the system having the N501K mutation, it shows the lowest distance between residue 501 and residue 383, 386, and 387 belonging to the ACE2 receptor. Extending the analysis to any other residue of the spike protein interface, we investigate the local organization of the intermolecular contacts through centrality measures. To this end, both the *Betweenness centrality* and *Closeness centrality* parameters (see Methods) were considered as a local descriptor of the interaction of each residue, since these are certainly two of the most widely used descriptors for the centrality analysis of a node[38] (See Fig. 5b).

In order to compare the different interaction organizations of each molecular system, we performed a clustering analysis (see Fig. 5c) considering a single vector for each system, which is composed of the combination of the two descriptors considered. The profile is relative only to the residues belonging to the spike interface (see Fig. 4a).

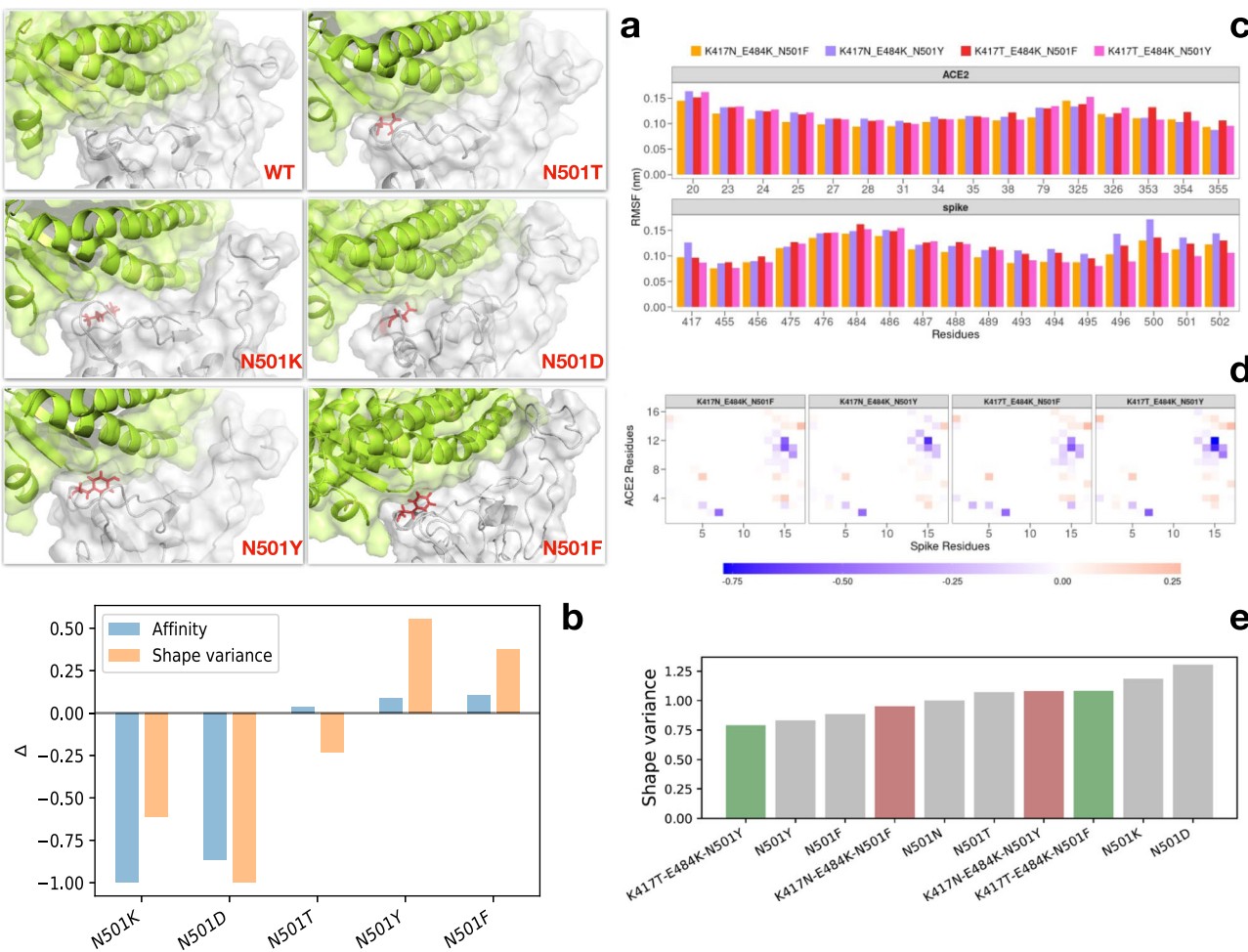

**Fig. 6 Analysis of the shape complementary of the binding regions. a** Cartoon representations of the binding region of the complex formed by SARS-CoV-2 spike protein (gray) and human ACE2 receptor (green) for the wild-type complex and the five single-mutation variants. The spike residue 501 is represented with red sticks. The molecular surfaces for both spike and ACE2 are shown. **b** Experimental affinity values (taken from ref. [27]) for the spike-ACE2 complexes (blue bars) and shape variance as measured by the Zernike descriptors on molecular dynamics configurations (orange bars). Both quantities are obtained by difference with respect to the reference complex data (see Table 1). **c** Root mean square fluctuations (RMSF) of ACE2 (top) and SARS-CoV-2 spike (bottom) protein residues found in interaction during the dynamics for the South African and Amazonian variants together with the two versions carrying phenylalanine at position 501. **d** Difference between the contact probability matrices of the interacting residues between each three-mutation spike variant and the Wuhan (WT) one (see Table 1). Residues are considered as interacting if the distance between their α-carbons is lower than 9 Å. **e** Shape variance as measured by the Zernike descriptors on molecular dynamics configurations for all the variants reported in Table 1.

Also in this analysis, we find a clear separation of mutations with lower binding affinity from the others, showing the ability of our molecular dynamics-based approach to distinguish between high and low-affinity systems. Interestingly, the WT form of the complex occupies an intermediate position between low-affinity and high-affinity mutations. In particular, the WT system is characterized by a profile very similar to the N501T mutation, which has the same amino acid in position 501 of the SARS-CoV system.

On the other side, the Amazonian variant and the South African variant appear to have a very similar organization of contacts with the positive control and the English variant, respectively. Interesting to note that the Amazonian variant, which is causing an important concern in the world, is very similar to the system characterized by the single mutation N501F, the one with the best experimental binding affinity. Similarly, the South African and English variants are in close proximity in terms of their Betweenness and Closeness properties, as they exhibit similar behavior in terms of infection transmission.

**Shape variation of binding site region.** Finally, we focused on the whole binding region of both SARS-CoV-2 spike protein and ACE2 receptor and assessed the shape complementarity of the binding site. To do so, we first expanded the molecular surfaces of the two molecular partners (see Fig. 6a) on the basis of the 3D Zernike moments (see Methods) and then computed the Euclidean distance between the two sets of invariant descriptors. We repeated the procedure for 250 configurations sampled from the equilibrium of each of the molecular dynamics simulations we performed. We thus obtained a distribution of Zernike distances, i.e., of complementarity scores, for each investigated variant. As one could expect, all distributions are centered around similar values of complementarity since the overall shape of the binding site does not undergo substantial conformational changes upon a few point mutations. However, looking at the variance of the distributions for the five single-mutation variants, we found a similar trend between the experimentally measured complex affinities and the variance of the shape complementarity[39]. This can be seen from Fig. 6b, where we compared the difference in

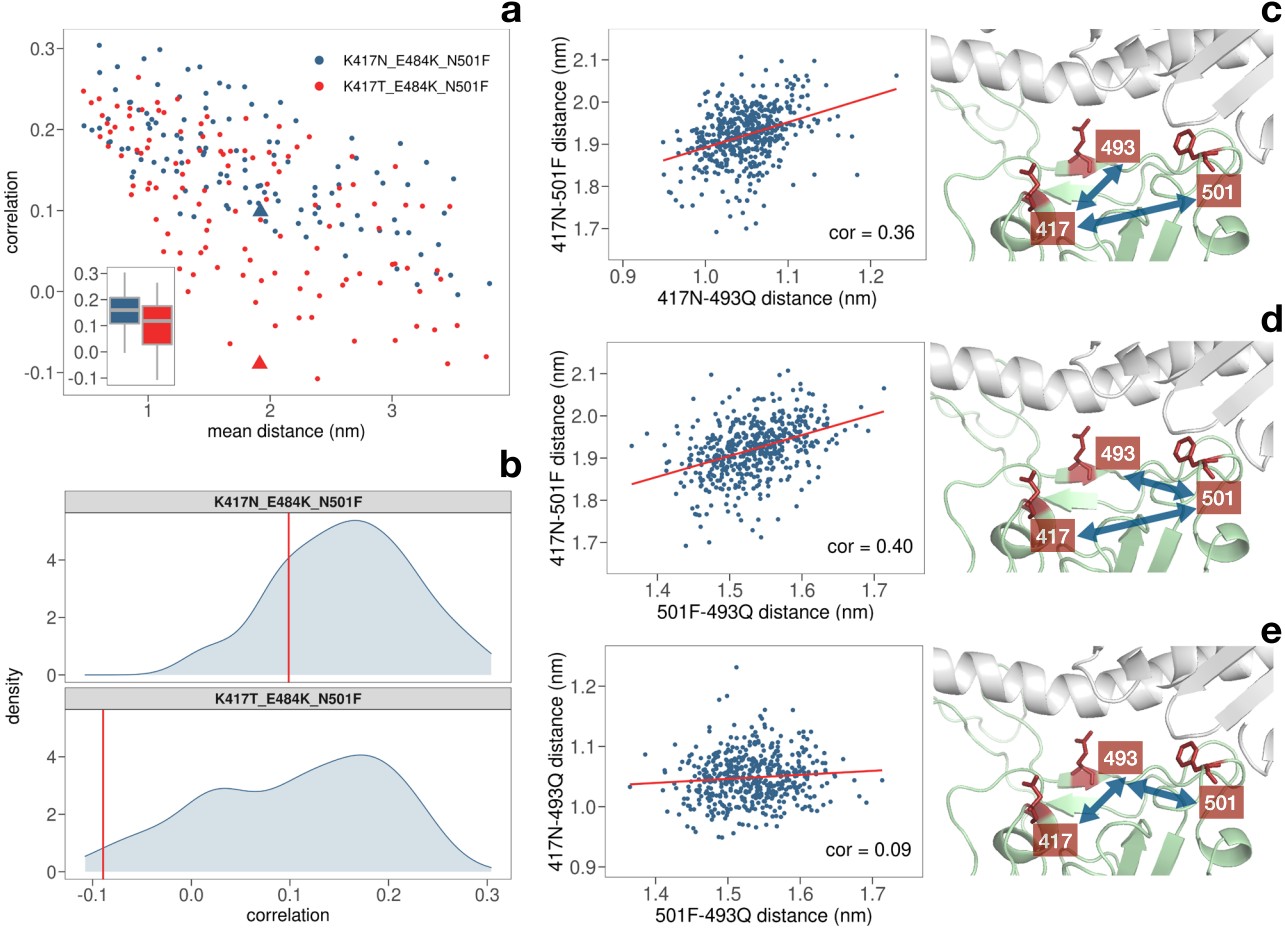

**Fig. 7 Analysis on the residues' motion in variants carrying the 501F mutation. a** Spatial correlation of the motion of all possible couples of residues of the spike protein as a function of their average spatial distance (of the residue centroids) during the simulation. Blue dots correspond to the residues of the South African variant with the putative N501F additional mutation with red dots represent all the couples of residues of the Amazonian variant with the N501F mutation. Box plots of the correlations are shown in the inset. **b** Distribution of the correlations in the motion of all the couples of residues having a sequence distance higher than three residues. Red vertical lines mark the correlation of residues 417–501. **c** Distance between the centroids of residues 417–501 vs resides 417–493 during the molecular dynamics simulation of the South African variant with the putative N501F additional mutation. Cartoon representation of the binding region between spike (green) and ACE2 receptor (gray), highlighting in red the key residues 417, 493, and 501. **d** Same as in (**c**) but for couples 417–501 vs 493–501. **e** Same as in (**c**) but for couples 417–493 vs 493–501.

affinity and shape variance of the five investigated variants with respect to the wild type. It is interesting to note that the N501Y variant displays a lower shape variance with respect to the wild type than the N501F variant. Unfortunately, as far as our knowledge goes, no binding affinity data are available for the three-mutation variants (i.e. the South African and Amazonian variants). We thus proceeded to compute the shape variance from the molecular dynamics configurations and compare them with single-mutation ones. Interestingly, we found that both variants show a lower variance with respect to the negative controls (N501D and N501K), with the South African variant being more motile than the English one while the Amazonian variant displaying the smallest variance (see Fig. 6c–d). This result provides an important example of the not-trivial effect of cooperativity. In fact, the same mutation (Y501F) gives an opposite outcome in the two variants, which now differ in terms of the interactions between a couple of residues 417T-501F against 417N-501F. Finally, we run two additional molecular dynamics simulations substituting amino acid F to Y in the two triple-mutated variants in order to check whether such mutation could bring an enhancement in the binding propensity. Looking again at the shape variance (see Fig. 6e), we observed that the modified South African and Amazonian variants behave oppositely: while a Y to

F mutation on the 501 residue increases the complementarity variability for the Amazonian variant thus worsening its binding capacity, the same mutation on the South African one brings a stabilization effect, with the complex maintaining a more stable shape complementarity during the dynamics.

**Analysis of motion between the key residues for the putative variants carrying the 501F mutation.** To better elucidate the non-trivial interplay between residue 501 and 417, we performed a correlation analysis between the motions of the binding site residues, both considering the case in which the mutation increases the stability (South African variant with additional 501F mutation) and the one where it decreases (Amazonian variant carrying the additional 501F mutation). Results are displayed in Fig. 7a.

Indeed, the average correlation of the residues belonging to the South African variant and the Amazon variant is 0.16 and 0.10 respectively. Notably, this substantial difference in motion properties of the whole spike binding region is caused by just one different residue in position 417.

We then focused on the 501–417 residue pair and their cooperative action. In Fig. 7b, we show the correlations of the

501–417 residue pair in the two systems with respect to the distribution of the correlations between all pairs of spike residues forming the binding region. In particular, only residues that are more than two residues apart in sequence have been selected to remove trivial correlation due to close proximity. Interestingly, the 501F–417N pair in the modified South African variant (predicted to be more stable) has a positive correlation of 0.10, while the 501F–417T residue pair in the modified Amazonian variant (predicted to give a less stable binding) has a negative correlation of −0.09.

Finally, we consider the correlated motion that the two residues have with the other residues of the binding region. Considering the distance between the centroids of the residue side chains, we compared the trend of the distance of each residue pair involving residue 417 and/or 501 as a function of time. Next, we looked for couples having a high correlation in the modified South African variant and a low correlation in the modified Amazonian system (see Supplementary Table I). Interestingly, only two pairs were selected and they both involve the same residue, i.e., residue Q493.

In particular, if one looks at the modified South African variant, the pair of residues 417–493 has a high Pearson correlation value (0.36) with the pair 417–5017, thus indicating a possible functional role (see Fig. 7c). Similarly, the 417–501 and 501–493 residue pairs display a Pearson correlation value of 0.40 as one can see from Fig. 7d. On the other hand, the pairs formed by residues 501 and 417 with residue 493 do not move in a correlated way (Fig. 7e), showing that residues 417 and 501 shares a direct correlation. Overall, our analyses add further residue-level evidence of the cooperative effect of residues 417–501 and suggest that residue 493 could be another important residue involved in the binding stability of the complex[40].

## Conclusions

The major goals of a virus are replication and spread, and SARS-CoV-2 coronavirus is constituting no exception. The vast, worldwide, diffusion of the SARS-CoV-2 epidemics indeed is providing the virus the possibility of exploring the genomic landscape and accumulating mutations. The resulting viral variants are subject to natural selection; thus, mutations that increase the diffusion rapidly get fixed in the viral genomic pool. Besides the transmission of the virus itself, antiviral treatments also contribute to selection. Indeed with the introduction of vaccines, new mutations inducing escape and resistance to available treatments are expected to couple with more virulent strains[41–43]. The new frontier for fighting the COVID-19 pandemic seems to become even more based on controlling and understanding the accumulation of variations in the SARS-CoV-2 genome.

To date, several different variants have already emerged. In particular, those with mutations on the spike protein are attracting a lot of attention as this protein is responsible for the binding to cellular receptor and attachment factors but also the primary target of the antibodies of the immune system.

Here, we deployed a set of molecular dynamics simulations to characterize the effect of different possible mutations of the spike residues 417, 484, and 501, which are the mutations found in the most relevant observed variants. In particular, mutation N501Y has been found in both the English, South African, and Amazonian variants. Analyzing the equilibrium configurations acquired by the spike-ACE2 complex in terms of residue fluctuations, networks of contacts, and conservation of shape complementary of the binding region, we found that indeed mutations on residue 501 strongly influence the dynamical stability of the complex. Most importantly, a phenylalanine substitution in position 501 increases the stability of the South

African variant via a cooperative action with residue 417N. In conclusion, our results suggest close surveillance for the emergence of such mutation to anticipate (and minimize) its effects in the viral spread and eventual early incorporation of this information into diagnostic and pharmacological procedures.

## Methods

**Structural data**. The complex of SARS-CoV-2 spike protein bound to the human ACE receptor has been taken from the PDB bank (PDB id: 6M0J). In particular, only the receptor-binding domain (RBD) of the spike and the extracellular domain of ACE2 are considered. Each variant considered in the present study has been obtained manually mutating the experimental complex via the Pymol software[35]. All information about amino acid substitutions is reported in Table 1.

Four additional structures were used to validate our procedure, i.e.,

- the spike protein in trimeric APO form of SARS-CoV-2 South African variant (PDB code: 7LYN);
- the spike protein in trimeric APO form of SARS-CoV-2 WT (PDB id:7KJ5);
- the experimental molecular complexes of the SARS-CoV-2 RBD bound to human ACE2 for the English variant (PDB id: 7MJN);
- and the experimental molecular complexes of the SARS-CoV-2 RBD bound to human ACE2 for the Amazonian variant (PDB id: 7NXC).

**Molecular dynamics simulations**. All simulations were performed using Gromacs[44]. Topologies of the system were built using the CHARMM-27 force field[45]. The protein was placed in a dodecahedric simulative box, with periodic boundary conditions, filled with TIP3P water molecules[46]. For all simulated systems, we checked that each atom of the proteins was at least at a distance of 1.1 nm from the box borders. Each system was then minimized with the steepest descent algorithm. Next, a relaxation of water molecules and thermalization of the system was run in NVT and NPT environments each for 0.1 ns at 2 fs time-step. The temperature was kept constant at 300 K with a v-rescale thermostat[47]; the final pressure was fixed at 1 bar with the Parrinello–Rahman barostat[48].

LINCS algorithm[49] was used to constraint bonds involving hydrogen atoms. A cut-off of 12 Å was imposed for the evaluation of short-range non-bonded interactions and the Particle Mesh Ewald method[50] for the long-range electrostatic interactions. The described procedure was used for all the performed simulations.

**Statistics and reproducibility**. All molecular dynamics simulations were 500 ns long, which guarantees that every system has reached equilibrium conformations. All subsequent analysis were performed sampling 500 frames from the equilibrium range.

**Molecular docking protocol**. Molecular docking between the unbound conformation of spike-RBD and ACE2 receptor has been performed using HDOCK software[51]. Apo form of RBD with K417N-E484K-N501Y mutations, is obtained by selecting the B chain from the trimeric form of the spike protein (PDB code: 7KJ5). Indeed, the B chain has the least number of missing residues with respect to the other two chains. The missing residues were modeled using SwissModel[52] in order to obtain the complete structure of a single chain of the spike protein. Subsequently, we performed the K417N, E484K, and N501Y mutations using the PyMol software[35]. Starting from this structure, we performed 500 ns of MD simulation for extrapolating the most representative structures through a clustering analysis on the first two principal components of the PCA. The clustering provided three representative structures, on which we performed the molecular docking procedure. On the other hand, the ACE2 receptor was selected from the spike-ACE2 complex (PDB code: 6M0J). Constrains on the binding residues were applied to molecular docking, selecting a large possible interaction region (116 residues and 194 residues for RBD and ACE2 interface respectively), using the known information on the binding regions.

**Interface residue definition and contact probability calculation**. For each analyzed complex, the interface was defined by taking ACE2 and SARS-CoV-2 spike residues whose α-carbons have a distance lower than 12 Å at time $t = 250$ ns, that is, after the equilibration phase. To end with a comparable set for all complexes, we selected the residues common to all interfaces obtaining 16 residuals for ACE2 and 17 for the spike protein. For each couple of interacting residues among the two proteins, we calculated the contact frequency, counting how many times each couple of residues had a distance lower than 9 Å between two α-carbons in each frame of the dynamics at equilibrium.

We got a 16 × 17 matrix for every complex reported in Table 1. Contact matrices shown in Fig. 3b are obtained by subtracting the wild-type one.

**Principal component analysis and clustering**. From the contact matrices of all complexes, we performed a principal component analysis (PCA) in which the starting matrix consisted of seven rows (the variants) and 272 columns (the contact frequencies of each pair of spike-ACE2 $\alpha$-carbons). The clustering analysis was performed using the "hclust" function of R, preserving the default clustering algorithm (the "complete" method). We first computed the contact percentage matrix for each system, as obtained from the molecular dynamics frames. For graphics analysis, we define contact between two residues if the distance between their center is less than 8.5 Å, as proposed in ref. [37]. Then, we defined a weighted graph for each matrix and calculated the betweenness centrality and closeness centrality parameters for each residue.

In particular, the betweenness centrality of a node $i$ is given by:

$$b_i = \sum_{k \neq i, l \neq i} \frac{s_{kl}(i)}{s_{kl}} \tag{1}$$

where $s_{kl}(i)$ is the number of weighted shortest paths ($s$) that go from node $k$ to node $l$ passing through node $i$; while $s_{kl}$ is the total number of weighted shortest paths from node $k$ to node $l$.

Similarly, the closeness centrality of a node $i$ is defined by the inverse of the average length ($<\cdot>$) of the weighted shortest paths to/from all the other nodes in the network:

$$c_i = \frac{1}{<s_{kl}>_{k \neq i, l \neq i}} \tag{2}$$

Both quantities were obtained via the corresponding functions of the "igraph" package of R[53].

For each system, we combined the two vectors (which have values of betweenness and closeness for single residues) normalized to 1. We used the Euclidean distance to compare every pair of vectors. Analyses were performed using R standard libraries[54].

**Molecular surface analysis via Zernike descriptors**. Given an ACE2-RBD complex simulated in molecular dynamics, for each protein, we calculate separately the molecular surface using DMS software[55]. Once extracted the portion of protein surface in interaction, with a voxelization procedure we represent the protein patch as a 3D function.

This 3D function can be described as a series expansion on the basis of the 3D Zernike Polynomials[28,56,57]. Taking the norm of the expansion coefficients we deal with an ordered set of numerical descriptors that compactly summarize the shape of the examined molecular surface.

Indeed, a function $f(r, \theta, \phi)$ can be written as:

$$f(r, \theta, \phi) = \sum_{n=0}^{\infty} \sum_{l=0}^{n} \sum_{m=-l}^{l} C_{nlm} Z_{nl}^m(r, \theta, \phi) \tag{3}$$

where $Z_{nl}^m$ are the 3D Zernike polynomials, while the coefficients $C_{nlm}$ are called Zernike moments.

The precision of the description can be selected by modifying the order of the expansion $N$. In this work, we fix $N = 20$, corresponding to 121 numerical descriptors representing each function.

The 3D Zernike Moments can be seen as:

$$C_{nlm} = \int_{|r| \leq 1} f(\mathbf{r}) \overline{Z_{nl}^m(r, \theta, \phi)} d\mathbf{r} \tag{4}$$

where $\overline{Z}$ represent the complex conjugate.

The norms of such moments, with respect to the index m, are invariant under translation and rotation. Indeed, the Zernike Descriptors are defined as:

$$D_{nl} = ||C_{nlm}|| = \sqrt{\sum_{m=-l}^{l} (C_{nlm})^2}. \tag{5}$$

The shape complementarity between two surfaces can be easily evaluated by applying a metric between the two vectors of numbers describing them[29,58,59]. Indeed, we adopted the euclidean distance. When two surfaces have a low distance between them, they are characterized by a similar shape and therefore they are suitable for binding.

**Reporting Summary**. Further information on research design is available in the Nature Research Reporting Summary linked to this article.

## Data availability
The data that support the findings of this study are available from the corresponding author upon reasonable request.

## Code availability
All codes used to produce the findings of this study are available from the corresponding author upon request. The code for the Zernike algorithm is available at https://github.com/matmi8/Zernike3D.

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

## Acknowledgements

The research leading to these results has been also supported by European Research Council Synergy grant ASTRA (n. 855923).

## Author contributions

M.M. performed the in silico mutations, carried out the statistical analyzes, and developed the numerical methods; M.M., L.D.R., L.B., and E.M. performed the molecular dynamics simulations and analyzed the data. G.P. and R.P. contributed additional ideas to the work and suggested biological tests with computational methods. A.B. and G.R. contributed with additional ideas and directed the computational choices on the basis of the biological and physical knowledge of the system; E.M. conceived the research. All authors wrote and revised the manuscript.

## Competing interests

The authors declare no competing interests.

## Additional information

**Peer Review Information** *Communications Biology* thanks the anonymous reviewers for their contribution to the peer review of this work. Primary Handling Editors: Karli Montague-Cardoso. Peer reviewer reports are available.

