## [Transparent Peer Review File · Communications Biology]

Reviewers' comments:

Reviewer #2 (Remarks to the Author):

In this manuscript, Miotto et al analyzed the effect of individual mutations in S protein on the overall affinity with ACE2 by performing extensive molecular dynamics simulations, particularly focusing on residue N501 of S protein. Overall analysis are solid and informative.

Comments

1. Binding of RBD of UK and South African strains to hACE2 are available in literature now, author may compare the simulation results with these newly available data.
2. Page 2, Reference related to N501F should be included here (reference 31).
3. Page 2, "6 mutated aminoacid" should be "amino acid".
4. Page 6, in RMSF analysis, UK strain (N501Y) seems to be the most stable (1.01Å), whereas South African strain appears to be the least stable (1.22Å). However, in the real world, South African strain is as transmissible as UK. Why?
5. Page 7, what is the reason using 8.5Å not 9Å here?

Reviewer #3 (Remarks to the Author):

The work by Miotto et al is aimed at identifying the effect of the amino-acidic mutation in the Spike protein of Sars-Cov-2 variants that allow the stabilisation of the binding with ACE2 receptor. In particular, the authors analyse three of the currently most spread variants, English, South African and Amazonian. The authors perform classical molecular dynamics simulation of the RBD-ACE2 complex of either WT (detected in Wuhan), the three variants, or different mutations at the level of residue N501.

Even if the work may provide some interesting information to better describe at molecular resolution the ability of the variants to increase the ability of the Sars-Cov-2 virus to achieve a higher infectivity, the work has a big limitation that in my opinion prevent its publication in this form.

Indeed, the authors analyse the stabilising effect of the mutation performing classical MD simulation starting from a conformation of the wild type RBD-ACE2 complex in which they introduce the amino acids changes. What is not taken in consideration by the authors is that the single or multiple mutations at the level of the RBD may change its structural profile and may allow it to acquire a different dynamics, followed by a different profile of interactions with the ACE2 receptor.

To characterise in depth the effect of the mutations in "stabilising" the interaction with the receptor, the authors should have simulated the three variants upon introduction of the mutation, analyse the structural-dynamical behaviour of the mutated RBD, extract the most representative conformations and dock those conformations against the ACE2 receptor to perform an exhaustive analysis of the RBD-ACE2 interaction profile and to extract the structural-dynamical effects of the mutants.

In other words, in order to study the effect of a point mutation on the binding between two proteins, it is not conceptually correct starting from the structure of the complex which is considered the WT. A mutated protein can sample a completely different conformational space with respect to the WT and, consequently, the binding mode could be completely different from that observed in the native condition, and thus the simulation of the protein-protein complex in which the starting point has the same coordinates of the WT but with just a single side chain changed could be considered a wrong starting point.

The work in this form cannot be accepted for publication since it has a big bias in the simulative conditions and the results proposed may be inconclusive.

Minor points

- 500ns cannot be considered "Extensive simulation" especially if the authors do not perform replicates.

- English wording needs a revision.

Dear Reviewers,

We would like to thank the Reviewers for their thoughtful comments and suggestions. Changes made in response to the points raised are reported in **red** if the text was removed and in **blue** if the text was added in the 'Diff.pdf' file. Our replies to the Reviewers' comments are given in the following in **blue**.

Thank you for your work.

Kind regards,

Edoardo Milanetti

for Mattia Miotto, Lorenzo Di Rienzo, Giorgio Gosti, Leonardo Bo', Giacomo Parisi, Roberta Piacentini, Alberto Boffi, and Giancarlo Ruocco

Reviewer #2

In this manuscript, Miotto et al analyzed the effect of individual mutations in S protein on the overall affinity with ACE2 by performing extensive molecular dynamics simulations, particularly focusing on residue N501 of S protein. Overall analysis are solid and informative.

We thank the Reviewer for the positive assessment of our work. We address all the points raised in detail below.

Comments

1. Binding of RBD of UK and South African stains to hACE2 are available in literature now, author may compare the simulation results with these newly available data.

Indeed, following the Reviewer's suggestion, we found that the experimental molecular complexes of the SARS-CoV-2 RBD bound to human ACE2 for the English (pdb id: 7mjn) and Amazonian (pdb id: 7nxc) variants are now available in the Protein Data Bank (PDB).

In addition, we selected a structure of the Spike in the unbound form (pdb id: 7lyn) and we used it to further check our mutational protocol. In particular, we performed molecular dynamics simulation of (i) the Spike-RBD-ACE2 complex (in which the mutations were computationally modeled starting from the experimental wt form of the complex); (ii) the spike RBD bound conformation (same as in point (i) but removing ACE2); (iii) the experimental APO form of the RBD of the South African variant (pdb: 7lyn) as directly suggested by Reviewer, and (iv) a computationally-obtained APO conformation extracted from the experimental wt APO conformation of the spike trimer, computationally mutated into the South African variant. The analysis of all these simulations confirmed that the obtained results are not significantly affected by the choice of a different simulation starting point, being it either an experimental structure or a computationally-mutated one. More specifically, the results suggest that the behaviour of the structure after mutation is directly linked to the effect of the mutation itself. Furthermore, a structural comparison shows that the structures sampled by MD of experimental structure are very similar, in terms of RMSD descriptor, to the structures obtained by MD of modelled spike structure. **We have now summarized and discussed the results of those analyses in the novel section 2 of the Results, in new Figure 2 and in the Supplementary Information.**

We finally would like to thank the Reviewer for notifying us the presence of the novel experimental structures.

2. Page 2, Reference related to N501F should be included here (reference 31).

Done.

3. Page 2, “6 mutated aminoacid” should be “amino acid”.

The Reviewer is right, we have now amended it.

4. Page 6, in RMSF analysis, UK strain (N501Y) seems to be the most stable (1.01A), whereas South African strain appears to be the least stable (1.22A). However, in the real world, South African strain is as transmissible as the UK. Why?

Indeed, the Reviewer raised an intriguing question. Some amino acid mutations near the interfaces have the ability to cause local conformational changes, producing high-affinity spike-ACE2 complexes, which are the basis for efficient transmission of viruses. However, the direct relationship between the binding affinity and the transmissibility is not trivial.

In this case, the binding affinities of these two strains appear to be quite comparable, but multitude of factors can influence the binding affinity, and the resulting transmissibility of the virus at the phenotypic level. Indeed, binding affinity is given by the combination of different microscopic components, like the molecular motility, the shape complementarity, and strength of the interaction between residues. In fact, in the present work, we tried to assess the stabilization role of the mutation by measuring and comparing some of those quantities, each capturing a part of the overall affinity. We note that the two mutations are most similar in terms of node Betweenness and Closeness, two network-based descriptors that quantify the compactness of the interaction residue network (see Figure 5b of the novel version of the manuscript). We have now added some comments in the manuscript in this respect.

5. Page 7, what is the reason for using 8.5A not 9A here?

We took the 8.5 A value from the work of Chakrabarty et al. [Ref 32 of the old manuscript], where similar network analyses were performed on protein structures. In particular, for this analysis we wanted to consider the side chain disposition, since in the residue-residue interaction network, it plays a crucial role. To this end, the centroid of any side chain is the most representative point for describing the side chain orientation. In Chakrabarty et al. paper, the best threshold for evaluating the centroid-centroid contact is discussed. We have now added a brief explanation in the manuscript and the citation to the reference work.

The work by Miotto et al is aimed at identifying the effect of the amino-acidic mutation in the Spike protein of Sars-Cov-2 variants that allow the stabilisation of the binding with ACE2 receptor. In particular, the authors analyse three of the currently most spread variants, English, South African and Amazonian. The authors perform classical molecular dynamics simulation of the RBD-ACE2 complex of either WT (detected in Wuhan), the three variants, or different mutations at the level of residue N501.

Even if the work may provide some interesting information to better describe at molecular resolution the ability of the variants to increase the ability of the Sars-Cov-2 virus to achieve a higher infectivity, the work has a big limitation that in my opinion prevent its publication in this form.

We thank the Reviewer for the appraisal of our work. Regarding the limitation, the Reviewer is concerned about, we have addressed this aspect both in the following discussion and in the revised version of the manuscript, where additional molecular dynamics simulations and analyses were carried out to validate our procedure according to the Reviewer comments.

Indeed, the authors analyse the stabilising effect of the mutation performing classical MD simulation starting from a conformation of the wild type RBD-ACE2 complex in which they introduce the amino acids changes. What is not taken in consideration by the authors is that the single or multiple mutations at the level of the RBD may change its structural profile and may allow it to acquire a different dynamics, followed by a different profile of interactions with the ACE2 receptor.

To characterise in depth the effect of the mutations in “stabilising” the interaction with the receptor, the authors should have simulated the three variants upon introduction of the mutation, analyse the structural-dynamical behaviour of the mutated RBD, extract the most representative conformations and dock those conformations against the ACE2 receptor to perform an exhaustive analysis of the RBD-ACE2 interaction profile and to extract the structural-dynamical effects of the mutants.

We see the Reviewer's point. Given the availability of the structure of the wild type complex, we have preferred to rely on the experimental structure and perform point mutations directly on the binding interface. However, we agree with the Reviewer that a test on the reliability of the sampled conformations during the simulation is necessary to validate the obtained results.

Given the importance of this analysis, we have discussed it in a novel section of the Result, where we show the validation of the approach we proposed in this work, for the studied system.

To this end, we performed two different kinds of analyzes:

1)

Following the suggestion of Reviewer 2, we exploited the recent availability of the spike-ACE2 experimental complexes of the English and Amazonian variants as a first, important validation. In fact, we performed a molecular

dynamics simulation for both the English and the South American variant's complexes, verifying that the configurations explored by the two simulations (which have an experimentally solved structure as starting structure) overlap with the structures sampled during the MD of the complexes obtained with our computational approach.

More specifically, we compared the structures obtained for the both simulations through a Principal Component Analysis (PCA). Indeed, each structure coming from the simulation generated by the experimental structure is projected into the essential space, defined by the two principal components calculated considering the trajectory of the complexes obtained with our approach. Therefore, in a compact and easily interpretable way, we check the overlap between the two sets of structures, which indeed have a high degree of similarity in terms of the backbone conformation projected into the first two principal components. **More technical details are in the Methods section and in the new section of the Results (see also new Figure 2).**

2)

Leveraging from Reviewer 2's request, we used the last of the three variants discussed in this work (the South African variant), for which no experimental complex structure is available, to check whether mutating the spike protein when in complex has a different effect on binding with the ACE2 receptor in dynamics, with respect to mutating the apo spike structure and then obtaining the complex via a docking procedure.

For this purpose, we compared the spike protein structures sampled in four different simulations:

- Spike-ACE2 complex originally in this work (in which the mutation of the variant was computationally designed directly at the binding interface of the experimental wt form of the complex).
- HOLO conformation of the spike protein obtained according to the procedure adopted in this work and then simulated free in solution.
- experimental APO form, considering as starting structure the pdb code: 7lyn
- computational APO conformation obtained by manual mutation of the experimental wt APO conformation (pdb id:7kj5): in this case we first design the mutation of the variant starting from the APO wt form and then we perform the simulation to investigate the adaptation of the structure following the mutation.

This last procedure follows the protocol proposed by the Reviewer, identifying the centroids of the clustering analysis of the sampled configurations and performing molecular docking with the ACE2 receptor structure. The results of this analysis show a similar behavior for the obtained configurations, suggesting that for this particular system, the two procedures may be interchangeable. **The results of these analyses have been added in the Results of the novel version of the manuscript, in the novel Figure 2 (that we also reported below) and in the Supporting Information.**

In other words, in order to study the effect of a point mutation on the binding between two proteins, it is not conceptually correct starting from the structure of the complex which is considered the WT. A mutated protein can sample a completely different conformational space with respect to the WT and, consequently, the binding mode could be completely different from that observed in the native condition, and thus the simulation of the protein-protein complex in which the starting point has the same coordinates of the WT but with just a single side chain changed could be considered a wrong starting point.

Although the Reviewer's comment is absolutely right, it must be pointed out that the particular system we investigated here, i.e the Spike-ACE2 complex and specifically the Spike RBD , appears to be quite stable to point mutations taking place on its binding site. This is testified both by the comparison between mutated vs experimental complexes of the English and Ammazzoian variants, and by the analyses on the MD of the South African variant's RBD in apo and mutated-holo forms. We might speculate that the peculiar stability of the RBD fold is motivated by the fact that the Spike protein has to continuously optimize its binding site to infect different hosts in different conditions; thus it could be optimal to have a stable fold upon single/few point mutations.

The work in this form cannot be accepted for publication since it has a big bias in the simulative conditions and the results proposed may be inconclusive.

We have now thoroughly revised the manuscript and hope to have addressed all the Reviewer concerns.

Minor points

- 500ns cannot be considered "Extensive simulation" especially if the authors do not perform replicates.

We see the Reviewer point. We have now modified the text accordingly.

- English wording needs a revision.

We have now thoroughly checked the text according to the Reviewer's suggestions. We hope readability has now improved.

Reviewers' comments:

Reviewer #1 (Remarks to the Author):

The manuscript by Milanetti et al uses molecular dynamic simulations to characterize the stabilizing effect of particular residue changes on a selected set of Spike variants, especially the residues 417, 484, and 501. They found the mutations on N501 influence the dynamical stability of the RBD-ACE2 complex. Furthermore, they also found that a phenylalanine substitution in position 501 increases the stability of the South African variant via a cooperative action with residue 417T. The paper is well organized and provides insight into the key features for the stability of the RBD-ACE2 complex and that may help to monitor further possible spike variants.

Major concern:

One of the major conclusions from the authors is about the N501F could cooperate with K417 to increase the Spike protein stability. However, in their data, they also found opposite outcomes in SA variant and BR variant when using the same substitution Y501F. Did the author have other data about the effects of residue K417 to further support this conclusion?

Minor points:

1. On page2, in the 'results and discussion' section, the author describes "... The English variant has the G614D mutation together with mutations/deletions ...". The UK variant has the D614G substitution together with others according to all other published data. Please check your data and correct it.
2. On page 5, line 8-11, "we analyze the South African and Amazonian variants carrying the mutations K417T-E484K-N501Y and K417N-E484K-N501Y, respectively". The SA variant has the K417N substitution and the Brazil variant has the K417T substitution. Please correct it.
- 3, on page 8, in the 'conclusion' section, the author wrote "... of the South African variant via a cooperative action with residue 417T. ", the SA variant contains K417N substitution, please correct it.

Reviewer #2 (Remarks to the Author):

This reviewer is satisfied with all changes that the authors made except for one thing. The "Spike" protein may change to "S" protein or "spike" protein.

Reviewer #3 (Remarks to the Author):

I want to thank the authors for the effort they put in addressing my concerns and suggestions. I see they were able to reply to all my comments and I now recommend the paper for publication in its current form.

Dear Reviewers,

We would like to thank the Reviewers for their thoughtful comments and suggestions. Changes made in response to the points raised are reported in **red** if the text was removed and in **blue** if the text was added in the 'Diff.pdf' file. Our replies to the Reviewers' comments are given in the following in **blue**.

Thank you for your work.

Kind regards,

Edoardo Milanetti

for Mattia Miotto, Lorenzo Di Rienzo, Giorgio Gosti, Leonardo Bo', Giacomo Parisi, Roberta Piacentini, Alberto Boffi, and Giancarlo Ruocco

Reviewer #1

The manuscript by Milanetti et al uses molecular dynamic simulations to characterize the stabilizing effect of particular residue changes on a selected set of Spike variants, especially the residues 417, 484, and 501. They found the mutations on N501 influence the dynamical stability of the RBD-ACE2 complex. Furthermore, they also found that a phenylalanine substitution in position 501 increases the stability of the South African variant via a cooperative action with residue 417T. The paper is well organized and provides insight into the key features for the stability of the RBD-ACE2 complex and that may help to monitor further possible spike variants.

We thank the Reviewer for the positive assessment of our work. Responses to all the comments are provided below.

Major concern:

One of the major conclusions from the authors is about the N501F could cooperate with K417 to increase the Spike protein stability. However, in their data, they also found opposite outcomes in SA variant and BR variant when using the same substitution Y501F. Did the author have other data about the effects of residue K417 to further support this conclusion?

We see the Reviewer's point. Indeed, the possible cooperative effect we found between residues 417 and 501 needs (and it is worthy of) more investigation. With this aim, we carried out additional analyses on the modified South African and Amazonian variants (those with the additional N501F mutation). In particular, we focused on the spike binding site residues motion during the complexes' molecular dynamics, with particular attention to residues 417 and 501.

Analyzing the correlations between the motion of couples of residues, we found that the more stable complex (the modified South African one) displays a higher average positive correlation with respect to the less stable one (mod Amazonian one).

In particular, the correlation between the couple 417 and 501 behaves oppositely in the two systems, i.e. the mod South African couple has a positive correlation and the mod Amazonian pair a negative one.

Next, we looked for couples having a high correlation in the modified South African variant and a low correlation in the modified Amazonian system. Interestingly, only two pairs were selected and they both involve residue Q493 together with either residue 417 or 501, suggesting a possible key role played by such residue.

Overall, our additional analyses provided further residue-level evidence of the cooperative effect of residues 417-501 and also identified residue 493 as another residue responsible for the increased binding stability of the modified South African complex.

We have now added an entirely new Section in Results and Discussions, including a new Figure (number 7) to explain and describe the additional analyses we performed.

We hope that in the revised version of the manuscript we addressed the Reviewer's concern. Indeed, we would like to thank the Reviewer for pointing out this aspect since we believe that the results of these further analyses increased the findings of our work.

Minor points:

- On page2, in the 'results and discussion' section, the author describes "... The English variant has the G614D mutation together with mutations/deletions ...". The UK variant has

the D614G substitution together with others according to all other published data. Please check your data and correct it.

The Reviewer is definitely right. We have now corrected the typos and thank the Reviewer for spotting it.

- On page 5, line 8-11, “we analyze the South African and Amazonian variants carrying the mutations K417T-E484K-N501Y and K417N-E484K-N501Y, respectively”. The SA variant has the K417N substitution and the Brazil variant has the K417T substitution. Please correct it.

Done.

- On page 8, in the ‘conclusion’ section, the author wrote “... of the South African variant via a cooperative action with residue 417T. ”, the SA variant contains K417N substitution, please correct it.

We apologize for the various typos in the variant mutations, we have now fixed them and again we would like to thank the Reviewer for the careful revision.

Reviewer #2

This reviewer is satisfied with all changes that the authors made except for one thing. The "Spike" protein may change to "S" protein or "spike" protein.

We are pleased to know that we managed to answer the Reviewer concerns in a satisfactory manner. In the novel version of the manuscript, we have now fixed the highlighted typos and would like to thank the Reviewer for the comments, which we believed helped improve our work.

Reviewer #3

I want to thank the authors for the effort they put in addressing my concerns and suggestions.

I see they were able to reply to all my comments and I now recommend the paper for publication in its current form.

We are pleased to know that our work has been appreciated and would like to thank the Reviewer for her/his comments, which we believed helped improve our findings.

REVIEWERS' COMMENTS:

Reviewer #1 (Remarks to the Author):

The authors have addressed my concerns, the manuscript is good for publication now.